# LucFlow: A method to measure Luciferase reporter expression in single cells

**Sunil Nooti** [ID][☯], **Madison Naylor**[☯], **Trevor Long, Brayden Groll, Manu**\*

Department of Biology, University of North Dakota, Grand Forks, ND, United States of America

☯ These authors contributed equally to this work.
\* manu.manu@und.edu

**Data Availability Statement:** All relevant data are within the manuscript and its Supporting Information files (S1–S13 Files).

**Funding:** This work was supported by the National Science Foundation, https://www.nsf.gov

## Abstract

Reporter assays, in which the expression of an inert protein is driven by gene regulatory elements such as promoters and enhancers, are a workhorse for investigating gene regulation. Techniques for measuring reporter gene expression vary from single-cell or single-molecule approaches having low throughput to bulk Luciferase assays that have high throughput. We developed a Luciferase Reporter Assay using Flow-Cytometry (LucFlow), which measures reporter expression in single cells immunostained for Luciferase. We optimized and tested LucFlow with a murine cell line that can be differentiated into neutrophils, into which promoter-reporter and enhancer-promoter-reporter constructs have been integrated in a site-specific manner. The single-cell measurements are comparable to bulk ones but we found that dead cells have no detectable Luciferase protein, so that bulk assays underestimate reporter expression. LucFlow is able to achieve a higher accuracy than bulk methods by excluding dead cells during flow cytometry. Prior to fixation and staining, the samples are spiked with stained cells that can be discriminated during flow cytometry and control for tube-to-tube variation in experimental conditions. Computing fold change relative to control cells allows LucFlow to achieve a high level of precision. LucFlow, therefore, enables the accurate and precise measurement of reporter expression in a high throughput manner.

## Introduction

The regulation of genes in space and time is the result of the combinatorial action of multiple transcription factors (TFs) binding to enhancers, also called *cis* regulatory modules (CRMs) [1–3]. Multiple TFs binding to enhancers interact with each other and nucleosomes through cooperative binding and competition to activate or repress genes in response to environmental stimuli [4]. Enhancer dysfunction has been implicated in congenital disorders, cancers, and common complex diseases such as Type II diabetes and obesity [5]. Genome-wide association studies have revealed that non-coding variants of importance are enriched at putative enhancer regions, and the results of ENCODE and modENCODE suggest that at least 80% of the human genome is actually involved with the regulation of the 1% of the coding genome [6]. Despite the importance of CRMs, there has been limited progress in decoding their regulatory logic. For example, a recent systematic survey of ENCODE-based enhancer predictions showed that only 26% of predicted regions had regulatory activity [7].

[1942471 to Manu]. The funders had no role in study design, data collection and analysis, decision to publish, or preparation of the manuscript.

**Competing interests:** The authors have declared that no competing interests exist.

Approaches to decoding the regulatory logic of enhancers vary from painstaking mutagenesis [8, 9]—enhancer bashing—to biochemical modeling [10–12] and machine learning [13]. Regardless of the approach, reporter assays are a necessary and central part of *cis* regulatory analysis. A reporter assay introduces a transgene in which the expression of a reporter gene, which produces an inert product, is driven by combinations of promoters and enhancers. Several strategies have been utilized for transgenesis [14] such as transient transfection, random integration by viral transduction, site-specific integration, and BAC recombineering. There are also many choices of reporter genes and methods for measuring gene expression available, ranging from Luciferase protein assayed by luminescence in bulk to microscopy-based methods such as single-molecule FISH [15] and the MS2-MCP system [16]. More recently, RNA sequencing (RNA-Seq) coupled to transient transfection of reporter libraries, known as massively parallel reporter assays (MPRAs; [13, 17]) has allowed the discovery and characterization of enhancers on the scale of the genome.

Measuring reporter activity quantitatively and reliably requires choosing a reporter and an assay methodology that help accomplish the experimental objectives. The sensitivity, dynamic range, temporal resolution, and throughput are some of the parameters that are considered when determining the methodology best suited to a particular experiment. Generally, there is a trade-off between the accuracy or precision and the throughput of the experiment.

In mammalian systems, the most commonly used reporter is the firefly Luciferase [12, 18] and expression is usually measured in bulk by Luciferase assay after transfecting the cells transiently with the reporter gene. The cells are co-transfected with a second reporter gene, usually *Renilla luciferase*, transcribed from a constitutive promoter to control for cell number and transfection efficiency, which vary considerably from experiment to experiment. While transient Luciferase assays have a high throughput, high sensitivity, and high dynamic range, they are not suitable for time series experiments and have relatively low precision [19, 20]. Stable integration of the reporter gene into the host cells' genomes ameliorates the transfection variability problem and allows time series experiments. However, cell number still does vary from sample to sample and the luminescence of the sample must be normalized to cell number. A common method for doing so is to estimate cell number by measuring the total protein concentration of the sample [21] under the assumption that total protein is invariant under the treatments of the experiment. This step reduces the sensitivity of the assay as the BCA or Bradford assay have relatively narrow dynamic range and low sensitivity [22]. The throughput is also lower than transient assays since it may take weeks or months to establish a transgenic cell line.

In contrast to bulk assays, methods that measure reporter expression at single-cell resolution circumvent the complications associated with estimating cell number and enable novel modes of interrogating gene regulation. Single-molecule fluorescent *in situ* hybridization (smFISH) allows the detection and counting of individual RNA transcript molecules [23–25] in individual fixed cells, while fluorescent proteins [26, 27] and the MS2-MCP system [28] enable the measurement of steady-state protein expression and the rate of transcription in real time in living cells respectively. While smFISH and the MS2-MCP system provide absolute quantification and are suitable for characterizing dynamics, their throughput is low since they require laborious Z-stack and time-lapse microscopy respectively followed by computationally intensive image analysis to quantify reporter expression. Fluorescent protein reporters lend themselves to high throughput data collection by flow cytometry of live cells but are poorly suited for characterizing gene expression dynamics. Chromophore formation occurs on the scale of minutes to hours [29] and fluorescent proteins have half lives on the scale of hours to days [30, 31].

Despite the availability of the aforementioned single-cell techniques, bulk Luciferase assays retain several advantages. Luciferase proteins have been optimized to have short half lives on

the scale of minutes to hours [32], making them suitable for reporting gene expression dynamics. Luciferase assays are highly sensitive and convenient to conduct in large scale. Furthermore, a large number of transient reporter vectors as well as stable transgenic cell lines have been developed in the past and Luciferase assays are still utilized widely. We sought to combine the accuracy and precision of single-cell methods with the rapid turnover, convenience, and high throughput of bulk Luciferase assays.

We developed *LucFlow*, which measures the intracellular abundance of Luciferase protein by immunofluorescent staining followed by flow cytometry. We utilized transgenic murine myeloid cell lines [33] bearing the *luciferase* gene under the control of the promoter and enhancers of *Cebpa*, which encodes CCAAT Enhancer Binding Protein, alpha, a transcription factor important for myeloid differentiation to optimize and validate LucFlow. We tested fixation and permeabilization reagents, blocking reagents, and different antibodies to identify conditions allowing the robust detection and measurement of Luciferase abundance. We also made provision to include internal control cells that are fixed, permeabilized, and stained together with sample cells but can be separated during flow cytometry to control for day-to-day and tube-to-tube experimental error and serve as a unit of Luciferase abundance. We show that dead cells have residual Luciferase expression that reduces the accuracy of bulk assays but does not impact LucFlow since dead cells are excluded from the analysis. We also demonstrate that including internal control cells increases the precision of the measurement three fold. Finally, we compare bulk assays and LucFlow in measuring the dynamics of reporter expression during *in vitro* neutrophil differentiation. While the temporal pattern of reporter expression is consistent between the two methods at a qualitative level, the LucFlow detects higher fold changes, demonstrating that it is more sensitive.

## Materials and methods

### PUER cell lines and cell culture

The construction of PUER-derived reporter cell lines is described elsewhere [33]. PUER cells were cultured and differentiated as described previously [12, 34]. Briefly, PUER cells were grown in complete Iscove's Modified Dulbecco's Glutamax medium (IMDM; Gibco, 12440061) supplemented with 10% FBS, 50$\mu$M $\beta$-mercaptoethanol, 5ng/ml IL3 (Peprotech, 213–13). PUER cells were differentiated into macrophages by adding 200nM 4-hydroxy-tamoxifen (OHT; Sigma, H7904–5MG). Cells were differentiated into neutrophils by replacing IL3 with 10ng/ml Granulocyte Colony Stimulating Factor (GCSF; Peprotech, 300–23) and inducing with 100nM 4-hydroxy-tamoxifen (OHT; Sigma, H7904–5MG) after 48 hours.

### Reporter assays

**Bulk reporter assays.** PUER cells were seeded at a density of $2.5 \times 10^5$ cells/mL in 6 replicates before being differentiated as described above. Cells were sampled by washing twice in PBS and lysed in Glo Lysis Buffer (Promega, E2661) for 5 min at RT. The lysate was cleared by centrifugation at 12,000 rpm at 4˚C for 5 min. The Luciferase assay was performed with 100 $\mu$L of the cleared lysate (Steady Glo reagent, Promega, E2510) according to the manufacturer's instructions. Total protein concentration was determined with the remaining 100 $\mu$L of the lysate using the BCA assay (Thermo Scientific, 23227) and a standard curve constructed with Albumin. Luminescence and absorbance were measured in a multimode plate reader (Beckman Coulter, DTX-880).

**Single-cell reporter assays.** Single-cell flow cytometry-based Luciferase measurements were done by intracellular immunostaining of Firefly Luciferase. Samples were stained without spiking for the optimization experiments. For all other experiments, 0.5–1 million cells of each

sample were spiked with $\sim 0.25$ million cells of either the Luc$^-$ PUER or the Promoter line stained with CFSE (eBioscience, 65–0850). The samples were stained with the LIVE/DEAD Fixable Violet Dead Cell Stain Kit (Invitrogen, L34955) to exclude dead cells. Cell were fixed and permeablized with BD Cytofix/Cytoperm Fixation and Permeabilization Solution (BD Biosciences, 554722) or Transcription Factor Buffer Set; BD Biosciences) according to the manufacturer's protocol. Cells were blocked with either Armenian Hamster $\alpha$-mouse CD16–2 antibody (clone 9E9; BD Biosciences) or rat $\alpha$-mouse CD16/CD32 antibody (clone 2.4G2; BD Biosciences). Cells were then stained for intracellular Firefly Luciferase with 10.75 ng/$\mu$L (1:75 dilution) $\alpha$-Luciferase rabbit monoclonal primary antibody (clone EPR17790; Abcam) or $\alpha$-Luciferase N-terminal rabbit monoclonal primary antibody (clone EPR17789; Abcam) at indicated concentrations. A polyclonal Goat Anti-Rabbit IgG H&L conjugated to Alexa Fluor 647 (Abcam, ab150083) was used as a secondary antibody at a final concentration of 1 $\mu$g/mL (1:2000 dilution). Fluorescence was recorded on a BD FACSymphony analyzer. Debris was excluded as events with low FSC-A and SSC-A. Clusters of cells were excluded as events having high FSC-A relative to FSC-H. Dead cells were excluded as events with high LIVE/DEAD violet fluorescence. Spiked control and sample cells were identified as CFSE$^+$ and CFSE$^-$ events respectively. Separation index (SI) was computed as

$$0.995 \frac{\mu_{\text{sample}} - \mu_{\text{PUER}}}{p_{\text{PUER}}^{84} - \mu_{\text{PUER}}},$$

where $\mu_{\text{sample}}$ and $\mu_{\text{PUER}}$ are the median fluorescence of the sample and Luc$^-$ PUER cells respectively, while $p_{\text{PUER}}^{84}$ is the 84th percentile of PUER fluorescence.

**Calculation of fold change relative to undifferentiated Promoter cells.** The procedure for calculating Luciferase expression in units of the expression of the undifferentiated Promoter depends on whether the sample was spiked with PUER cells or with undifferentiated Promoter cells as internal control. In either case the fold change is the ratio of the background-subtracted fluorescence of the sample to that of the undifferentiated Promoter.

**Sample spiked with PUER cells.** PUER cells do not express Luciferase protein and are used to estimate background fluorescence. Each sample is spiked with PUER internal control cells, so that the background-subtracted signal is simply the difference between sample and PUER fluorescence. Let $f_{\text{sample}}^1$ and $f_{\text{sample}}^2$ be the median Luciferase fluorescence of the undifferentiated Promoter and the sample respectively and $f_{\text{control}}^1$ and $f_{\text{control}}^2$ be the median Luciferase fluorescence of the PUER internal control spiked into the corresponding samples. The Luciferase expression is the ratio of background-subtracted signals,

$$L_{\text{sample}} = \frac{f_{\text{sample}}^2 - f_{\text{control}}^2}{f_{\text{sample}}^1 - f_{\text{control}}^1}.$$

**Sample spiked with undifferentiated Promoter cells.** In order to estimate background, PUER cells must be spiked with undifferentiated Promoter cells and recorded. Let $f_{\text{sample}}$ and $f_{\text{sample}}^s$ be the median Luciferase fluorescence of the stained Luciferase protein in the sample and the spiked Promoter internal standard respectively. Also, let $f_{\text{PUER}}$ and $f_{\text{PUER}}^s$ be the median Luciferase fluorescence in PUER cells and Promoter internal standard they were spiked with respectively. The background fluorescence in the sample was estimated as

$$b_{\text{sample}} = f_{\text{PUER}} \frac{f_{\text{sample}}^s}{f_{\text{PUER}}^s}.$$

The Luciferase expression in the sample was then computed as

$$L_{\text{sample}} = \frac{f_{\text{sample}} - b_{\text{sample}}}{f^s_{\text{sample}} - b_{\text{sample}}}.$$

# Results

## Optimization of intracellular staining conditions

As a first step we found optimal conditions for intracellular staining that maximize the separation index (SI; see Methods) between cells that express Luciferase and those that do not. PUER cells are IL3-dependent bone marrow progenitors from PU.1$^{-/-}$ mice into which a 4-hydroxytamoxifen (OHT) inducible PU.1 transgene has been reintroduced and can be differentiated into neutrophils by OHT induction in GCSF-conditioned media (Fig 1B; [11, 12, 34]). We had previously created transgenic PUER cell lines that express Luciferase by knocking several reporter genes into the ROSA26 safe harbor site using CRISPR/Cas9 (Fig 1C; [33]). The reporters' expression is controlled by the promoter, either alone or in combination with different enhancers, of *Cebpa*, which encodes a transcription factor (TF) necessary for neutrophil development and is upregulated during differentiation (Fig 1A; [11, 12, 33]). In what follows, we refer to the unedited Luc$^-$ cell line as PUER, the line expressing Luciferase under the lone control of the *Cebpa* promoter as Promoter, and the lines bearing the CRM 7, CRM 16, and CRM 18 enhancers in addition to the promoter by their enhancer names (Fig 1C). We maximized the separation index (SI) of the Promoter line relative to the Luc$^-$ PUER line to determine optimal staining conditions (Fig 1A).

Intracellular staining of proteins involves the steps of fixation and permeabilization, blocking, and antibody staining. Although many fixation and permeabilization buffers are available commercially from manufacturers [35], they can be divided into two broad types. Methanol or detergents such as Triton permeabilize the nuclear membrane and might be necessary for staining nuclear proteins, but can disrupt them. Saponin or weaker detergents such as digitonin are gentler, but are mainly suitable for cytoplasmic proteins. Saponin permeabilization is transient unless saponin is present in the wash and storage buffers. We tested the BD TF buffer set (Fig 1D), which contains formaldehyde and methanol for fixation and permeabilization respectively, and the BD Cytofix/Cytoperm kit (Fig 1F), which contains formaldehyde and saponin for fixation and permeabilization respectively. Saponin-based permeabilization had a much higher SI than the methanol-based method (Fig 1D and 1F).

Having determined the optimal permeabilization conditions, we tested two different primary $\alpha$-Luciferase antibodies, a rabbit monoclonal (clone EPR17790; Abcam; Fig 1E) and an N-terminal specific rabbit monoclonal (clone EPR17789; Abcam; Fig 1F). The N-terminal antibody clone EPR17789 performed much better even though it was used at a 4.5-fold lower concentration than the other antibody. PUER cells express Fc gamma receptors (Fc$\gamma$R; [36]) which can result in significant non-specific staining and therefore must be blocked to reduce background. The most common reagent for blocking Fc$\gamma$R is the antibody against Fc$\gamma$RIII/II (CD16/CD32) but it was recently shown that many immune cells also express Fc$\gamma$RIV (CD16–2; [37]). We tested blocking using rat anti-mouse CD16/CD32 antibody (Fig 1F) vs Armenian Hamster anti-mouse CD16–2 antibody (Fig 1G) and found that the latter was superior to the former. Finally, we titrated the primary antibody to find the optimal concentration (Fig 1H). The optimal conditions utilized saponin-based permeabilization, the N-terminal specific primary antibody at a concentration of 2.37 $\mu$g/mL to 5.93 $\mu$g/mL, and blocking with the $\alpha$-CD16–2 antibody and achieved an SI of $\sim$4 (Fig 1G and 1H).

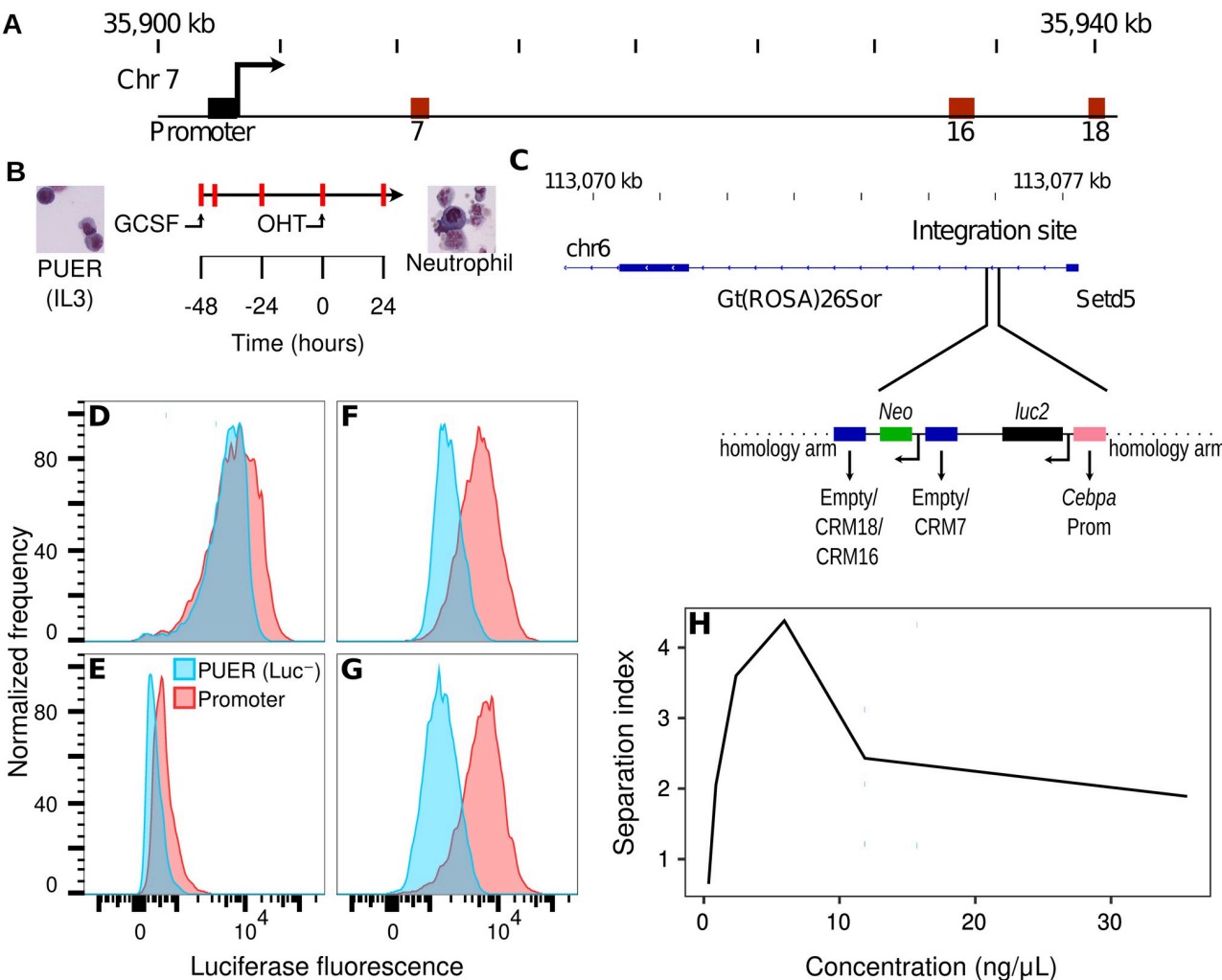

**Fig 1. Reporter design and the optimization of the immunofluorescent staining. A**. A 40kb region of chromosome 7 showing the *Cebpa* transcription start site (TSS), promoter, and CRMs (enhancers) 7, 16, and 18. **B**. PUER cells may be differentiated into neutrophils by substituting GCSF for IL3 48 hours prior to OHT induction. The red marks indicate timepoints at which Luciferase assays were conducted. **C**. Site-specific knock in of Luciferase reporters into intron 1 of the ROSA26 locus by CRISPR/Cas9 HDR. *luc2* gene expression was driven by the *Cebpa* promoter (orange) alone or along with CRM 7, 16, or 18. **D–G**. Histograms of Luciferase fluorescence measured by flow cytometry under different staining conditions. Cells carrying the reporter gene expressing *luc2* under the control of the *Cebpa* promoter in the ROSA26 locus (Promoter) are shown in red. Unedited cells lacking *luc2* (Luc⁻) are shown in blue. **D**. Fixation and permeabilization by formaldehyde and methanol (Transcription Factor Buffer Set; BD Biosciences), blocking by rat $\alpha$-mouse CD16/CD32 antibody (clone 2.4G2; BD Biosciences), and staining by 2.37 ng/$\mu$L (1:75 dilution) $\alpha$-Luciferase N-terminal rabbit monoclonal primary antibody (clone EPR17789; Abcam). **E**. Fixation and permeabilization by formaldehyde and saponin (Cytofix/Cytoperm; BD Biosciences), blocking by rat $\alpha$-mouse CD16/CD32 antibody (clone 2.4G2; BD Biosciences), and staining by 10.75 ng/$\mu$L (1:75 dilution) $\alpha$-Luciferase rabbit monoclonal primary antibody (clone EPR17790; Abcam). **F**. Fixation and permeabilization by formaldehyde and saponin (Cytofix/Cytoperm; BD Biosciences), blocking by rat $\alpha$-mouse CD16/CD32 antibody (clone 2.4G2; BD Biosciences), and staining by 2.37 ng/$\mu$L (1:75 dilution) $\alpha$-Luciferase N-terminal rabbit monoclonal primary antibody (EPR17789; Abcam). **G**. Fixation and permeabilization by formaldehyde and saponin (Cytofix/Cytoperm; BD Biosciences), blocking by Armenian Hamster $\alpha$-mouse CD16–2 antibody (clone 9E9; BD Biosciences), and staining by 2.37 ng/$\mu$L (1:75 dilution) $\alpha$-Luciferase N-terminal rabbit monoclonal primary antibody (clone EPR17789; Abcam). In all cases, a polyclonal Goat $\alpha$-Rabbit IgG H&L secondary antibody conjugated to Alexa Fluor 647 was utilized at a final concentration of 1 $\mu$g/mL (1:2000 dilution). **H**. Titration of the $\alpha$-Luciferase N-terminal rabbit monoclonal primary antibody concentration.

## Discriminating between live and dead cells is necessary for accuracy

Dead or dying cells often have high autofluorescence that can result in false positives. It is common practice to exclude dead cells from flow cytometric analysis based on the principle

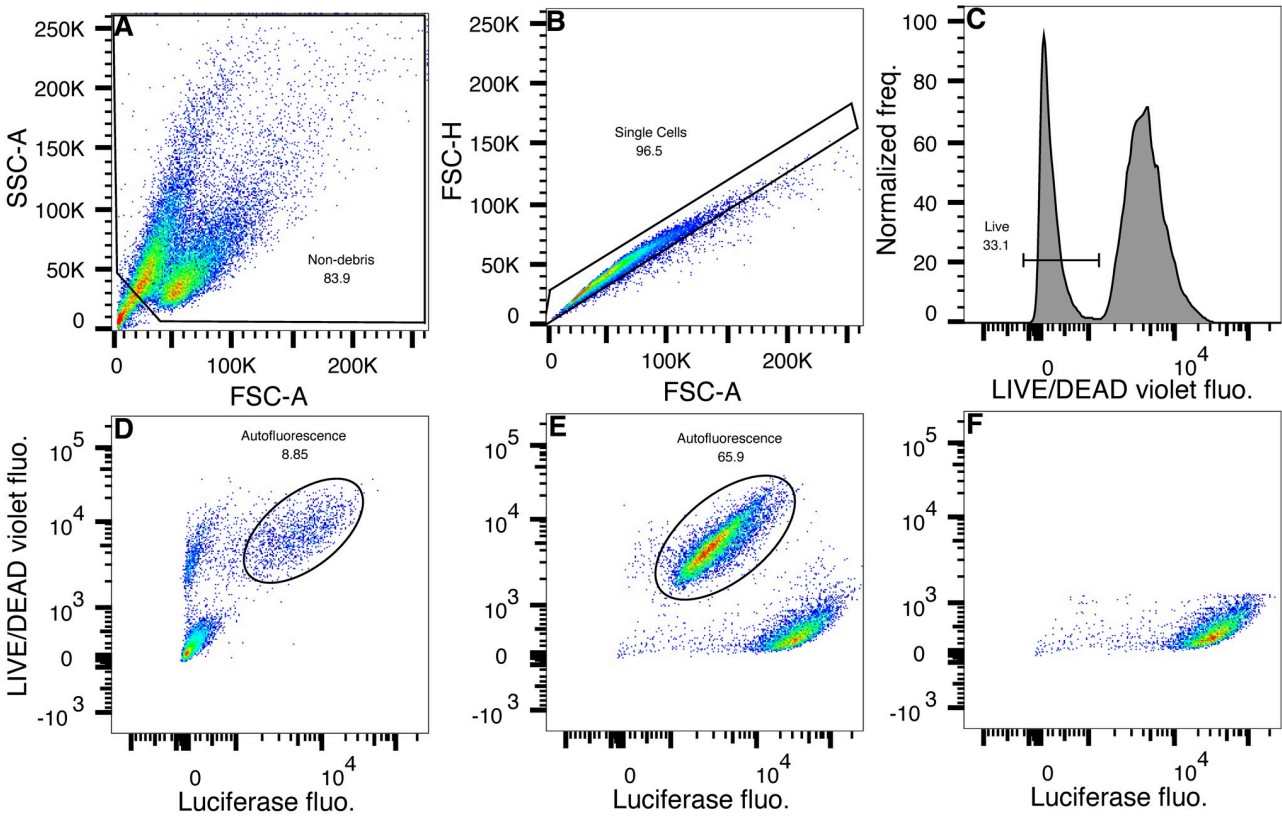

**Fig 2. Dead cells autofluoresce but do not stain for Luciferase protein. A–C** Gating scheme. **A**. Debris was excluded as events with low forward (FSC-A) and low side scatter (SSC-A). **B**. Singlets were identified by proportionality between forward scatter area (FSC-A) and height (FSC-H). **C**. Dead cells were excluded based on the fluorescence of a fixable viability stain, LIVE/DEAD Fixable Violet. **D–F**. Dot-plots of Luciferase and LIVE/DEAD Violet fluorescence. **D**. Luc⁻ PUER cells stained only with the viability dye. The singlet subpopulation has been plotted with Luciferase fluorescence on the *x*-axis and LIVE/DEAD Violet fluorescence on the *y*-axis. Autofluorescent dead cells are highlighted with an ellipse. **E**. CRM 18 cells stained for viability and Luciferase. The singlet subpopulation has been plotted with Luciferase fluorescence on the *x*-axis and LIVE/DEAD Violet fluorescence on the *y*-axis. Dead cells are highlighted with an ellipse. **F**. Live CRM 18 singlets gated as shown in panel C.

that DNA-binding or amine-reactive dyes [38] can penetrate the compromised cell membranes of dead cells to stain them but cannot do so in live cells. We inspected the fluorescence in the Luciferase channel of unstained PUER and stained CRM 18 cells, which express Luciferase under the control of the *Cebpa* promoter and CRM 18 (Fig 1C), by the standard gating scheme of excluding debris (Fig 2A) and clumped cells (Fig 2B). Stained Luciferase-expressing cells have two sub-populations, live cells that have high Luciferase signal and dead cells that have low to intermediate Luciferase signal (Fig 2E). Unstained PUER cells also have two sub-populations, a live population with low Luciferase signal and dead cells with low to intermediate Luciferase signal, showing that the latter autofluoresce. Notably, the fluorescence of stained dead cells is at similar levels as the autofluorescence of unstained dead cells (Fig 2D and 2E; ellipses), demonstrating that dead Luciferase-expressing cells have no functional Luciferase protein. This implies that bulk assays, which cannot distinguish between dead and live cells, significantly underestimate Luciferase expression whereas our single-cell method is expected to yield a higher and more accurate estimate by excluding dead cells (Fig 2C and 2F). The exclusion of dead cells is highly important for accurate estimation of reporter activity during differentiation since it causes a high level of cell death and viability can be as low as 50% [34].

## Inclusion of an internal control increases precision

While flow cytometry is quite robust at distinguishing between cell types when there is a large difference in expression levels, measuring expression at a quantitative level can be quite challenging. Median fluorescent intensity (MFI) is notorious for being highly variable from sample to sample where the coefficient of variation (CV) can be 30% or more [39]. The variability of MFI can be attributed to variation in experimental factors such as culture conditions, fixation and permeabilization, pipetting, cell number, cell viability, flow cytometer laser power, and compensation. The sample-to-sample variation contributed by flow cytometry can be controlled by computing the separation index between the sample and unstained or stained negative control, but the remaining variation stemming from the staining procedure is still significant.

In order to control for sample-to-sample variation caused by the experimental procedures, we spiked the sample's Luciferase-expressing cells with 10%–25% of Luc⁻ PUER cells. The PUER cells were labeled with carboxyfluorescein succinimidyl ester (CFSE) prior to spiking and served as an internal control for fixation, permeabilization, and staining procedures since the sample and control cells are treated equivalently. The sample and the internal control were deconvolved in flow cytometry by gating on CFSE fluorescence (Fig 3A and 3B) to reveal the Luciferase fluorescence distributions of the samples (Fig 3C and 3D). The CV of the MFI of various samples without normalization to spiked controls ranged from 59–105% (Fig 4). The SI relative to the internal control had considerably lower variation than MFI, with CV ranging between 27–37%. Finally, we computed the fold change in background-subtracted fluorescence between the sample and undifferentiated Promoter (see Methods and Fig 3C and 3D). The fold change had an even lower CV of 8–12%, demonstrating that our method can achieve a high level of precision. It is also possible to use Luciferase-expressing cells, such as undifferentiated Promoter cells, as controls although the procedure for computing fold change is slightly different (see Methods).

## Comparison of bulk vs single cell assays

We next compared the performance of LucFlow to bulk Luciferase assays by measuring the fold change of Promoter and CRM 18 cells during *in vitro* neutrophil differentiation. Differentiation was elicited by substituting GCSF for IL3, culturing in GCSF for 48 hours, and activating the PU.1 transgene by adding OHT [11, 12, 33, 34]. OHT addition is regarded as the 0h timepoint so that GCSF treatment begins at −48h. Reporter expression was measured at five time points and normalized to the expression of the undifferentiated (−48h) Promoter cells.

The temporal pattern of fold change is consistent between the two methods (Fig 5A); both the reporter genes exhibit early transient upregulation followed by a decline. The CRM 18 reporter always has higher expression than the Promoter reporter, demonstrating the enhancing activity of CRM 18. The measurements are correlated between the two methods ($r^2 = 0.75$; Fig 5B). Interestingly, the single-cell method consistently detects a higher fold change than the bulk one (Fig 5A and 5B). The ratio of the single-cell measurement to the bulk one is highest at the 0h and 24h timepoints, when there is a significant amount of cell death in response to GCSF treatment [34]. This suggests that the bulk measurements are confounded by the presence of dead or dying cells (Fig 2E). In contrast, LucFlow is robust to the presence of dead cells since they are removed from the analysis with a LIVE/DEAD stain (Fig 2C and 2F).

## Discussion

Gene expression, especially of developmental genes, is dynamically regulated in a complex scheme that involves multiple enhancers [3, 40–42] bound by multiple TFs [9, 43–45].

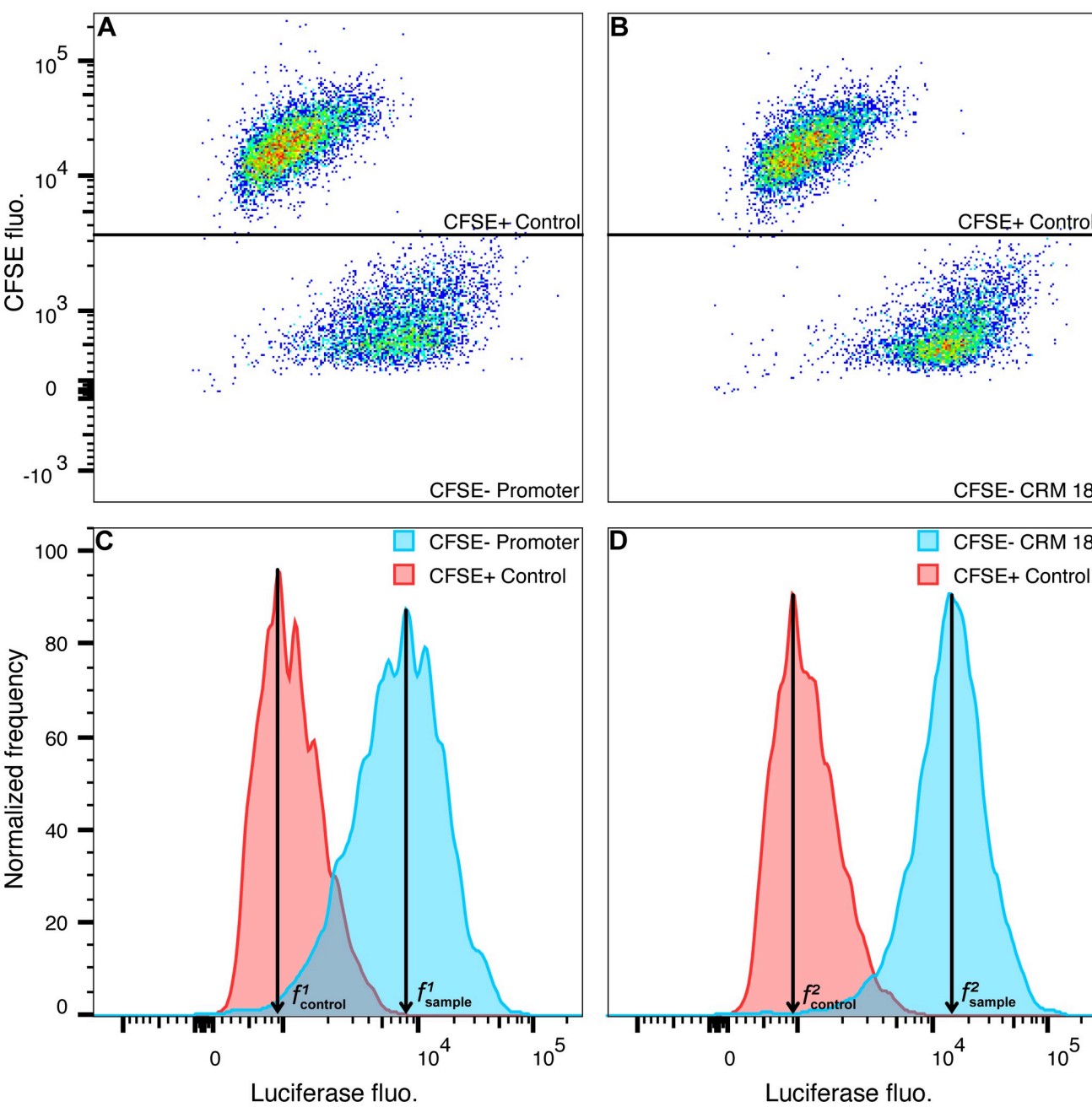

**Fig 3. Deconvolution of the sample and internal control and background subtraction. A,B**. Dot plots of Luciferase fluorescence vs. CFSE fluorescence. Gates separate the CFSE$^+$, Luc$^-$ PUER internal control cells from the CFSE$^-$ sample cells. **C,D**. Histograms of Luciferase fluorescence. The background-subtracted signal was determined by subtracting the median fluorescence of the control ($f_{control}^i$) from that of the sample ($f_{sample}^i$). **A,C**. The sample is the undifferentiated Promoter line. **B,D**. The sample is the CRM 18 line.

Reporter assays are an essential tool for studying gene regulation since they help reduce the complexity of the problem by measuring the expression of individual isolated enhancers or their combinations. While many techniques, ranging from bulk Luciferase assays to smFISH, are available for conducting reporter assays they involve a trade-off between accuracy and throughput. We developed a technique, LucFlow, which measures Luciferase reporter expression in single cells with flow cytometry. We have demonstrated here that LucFlow is more

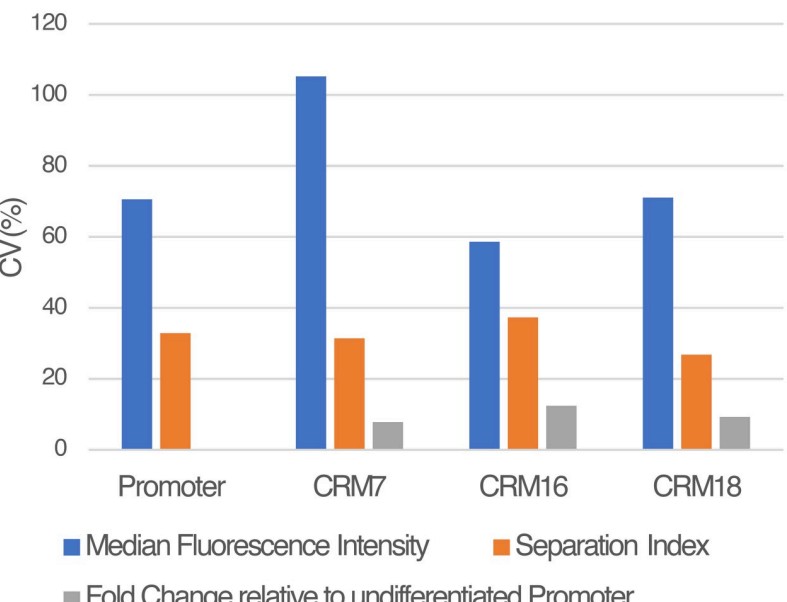

**Fig 4. Spiking internal control and fold change measurement increase precision.** The coefficient of variation (CV) of median fluorescence intensity (MFI), separation index (SI) relative to Luc$^-$ PUER internal control, and fold change relative to the undifferentiated Promoter cells are plotted on the *y*-axis.

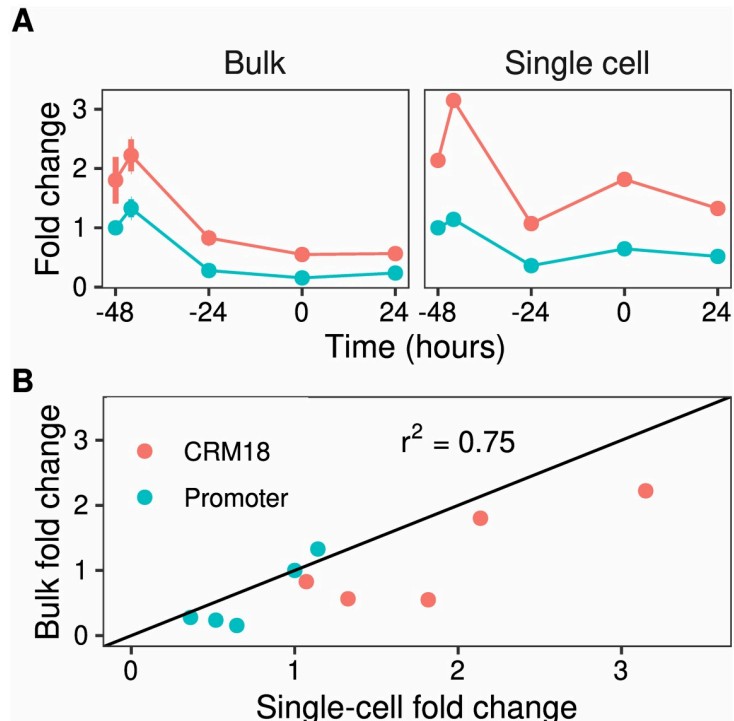

**Fig 5. Comparison of single-cell and bulk reporter assays. A**. Time series of Promoter and CRM 18 Luciferase reporter expression measured either in bulk with Luciferase assay or at a single-cell level with LucFlow. Cells were treated with GCSF at -48 hrs and OHT was added at 0 hrs. Reporter expression is plotted as the fold change relative to the expression of undifferentiated Promoter cells, prior to GCSF treatment. **B**. Scatter plot of the fold changes measured by the two methods.

accurate than bulk assays and has a high level of precision, while retaining sufficient through-put to conduct high temporal resolution time-series experiments.

In the course of optimizing and testing LucFlow we discovered that dead-cell contamination could result in inaccurate measurements in bulk assays. The signal from dead cells stained for Luciferase is indistinguishable from their autofluorescence (Fig 2D and 2E), which implies that dead cells do not contribute functional Luciferase protein to the cell lysate in bulk assays. This is consistent with the observation that increasing cell death results in lower Luciferase activity. For example, Mitsuki *et al.* [46] show that increasing DMSO-induced cytotoxicity results in a proportional reduction of Luciferase activity. Similarly, Didiot *et al.* [47] documented a reduction in Luciferase activity as cell viability is lowered with increasing antibiotic concentration. A lack of functional Luciferase protein in the dead cells of a sample would result in an underestimation of reporter expression in bulk assays if the sample were to have a significant proportion of dead cells. Dead cells are easily accounted for in flow cytometry by excluding them using a viability dye. Indeed, we observed higher fold changes in reporter expression with LucFlow compared to bulk Luciferase assays during differentiation (Fig 5), which causes significant cell death [34].

While it is generally appreciated that reporter assays should be conducted on healthy viable cells, this is not feasible in many situations, for example in differentiation experiments, when experimental treatments such as transfection cause mortality, or when conducting assays with primary cells. LucFlow should provide more accurate results in such experiments. Furthermore, although we have inferred the deleterious effect of dead cell contamination with Luciferase reporters, it likely also impacts other bulk reporter assays such as MPRAs [17]. Our results imply that even if a single-cell assay is not feasible, sorting cells by viability or removing dead cells using Annexin V antibody [48] prior to a bulk assay could improve accuracy.

LucFlow was able to achieve a CV of $\sim 10\%$ by spiking the sample with control cells stained with a dye that may be distinguished from sample cells during flow cytometry. The spiked cells undergo all experimental treatments, from fixation to flow cytometry in the same tube as the sample cells and control for tube-to-tube variation in the experimental treatment. The internal control cells can be either Luc$^-$ cells so they only have non-specific background staining or appropriately chosen reference Luc$^+$ cells, such as the promoter-only line utilized here. The former spiking scheme allows within tube background subtraction while needing an estimate of the fluorescence of reference cells from a separate tube to compute fold change, while the latter computes fold change within the same tube but estimates background from a separate tube. We have tried both approaches and they appear to perform equally well (Figs 4 and 5).

Luciferase reporters have been used extensively in the past and, despite the availability of single-cell and single-molecule methods, are still employed often due to their ease and high throughput. There are a large number of Luciferase-expressing lines available and LucFlow is compatible with transient assays as well. We anticipate that LucFlow could have broad applicability in studies of gene regulation by improving the accuracy and precision of Luciferase assays.

## Supporting information

**S1 File. Flow Cytometry Standard (FCS) file containing data from the Promoter line plotted in Fig 1D.**
(FCS)

**S2 File. FCS file containing data from the PUER line plotted in Fig 1D.**
(FCS)

**S3 File. FCS file containing data from the Promoter line plotted in Fig 1E.**
(FCS)

**S4 File. FCS file containing data from the PUER line plotted in Fig 1E.**
(FCS)

**S5 File. FCS file containing data from the Promoter line plotted in Fig 1F.**
(FCS)

**S6 File. FCS file containing data from the PUER line plotted in Fig 1F.**
(FCS)

**S7 File. FCS file containing data plotted in Fig 1G.**
(FCS)

**S8 File. FCS file containing data plotted in Fig 2.**
(FCS)

**S9 File. FCS file containing data used to plot Fig 3A and 3C.**
(FCS)

**S10 File. FCS file containing data used to plot Fig 3B and 3D.**
(FCS)

**S11 File. The data underlying the coefficients of variation plotted in Fig 4.** The CRM field
indicates which construct was assayed and Replicate is biological replicate number. The next
three columns are median fluorescence intensity, separation index, and fold change relative to
the undifferentiated Promoter cells.
(CSV)

**S12 File. Bulk luminescence data underlying the means and error bars plotted in Fig 5.** The
CRM field indicates which construct was assayed, Replicate is biological replicate number,
Time is the time point in hours during differentiation, and the Luminescence field is the mea-
sured bulk luminescence.
(CSV)

**S13 File. Flow cytometry data underlying the means and error bars plotted in Fig 5.** The
ControlLucFluorescenceMedian and SampleLucFluorescenceMedian fields are the median
fluorescence of internal control and sample respectively.
(CSV)

## Acknowledgments

We thank North Dakota Flow Cytometry & Cell Sorting Core for services.

## Author Contributions

**Conceptualization:** Sunil Nooti, Madison Naylor, Trevor Long,  Manu.

**Data curation:** Sunil Nooti, Madison Naylor, Brayden Groll,  Manu.

**Funding acquisition:**  Manu.

**Investigation:** Sunil Nooti, Madison Naylor, Brayden Groll.

**Methodology:** Sunil Nooti, Madison Naylor, Brayden Groll.

**Project administration:**  Manu.

**Resources:** Manu.

**Supervision:** Manu.

**Validation:** Sunil Nooti, Madison Naylor.

**Writing – original draft:** Sunil Nooti, Trevor Long, Manu.

**Writing – review & editing:** Sunil Nooti, Trevor Long, Manu.

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
