## [Decision Letter · Decision Letter 0]

15 Aug 2023

PONE-D-23-20181LucFlow: A method to measure Luciferase reporter expression in single cellsPLOS ONE

Dear Dr. Manu,

Thank you for submitting your manuscript to PLOS ONE. After careful consideration, we feel that it has merit but does not fully meet PLOS ONE’s publication criteria as it currently stands. Therefore, we invite you to submit a revised version of the manuscript that addresses the points raised during the review process.

We look forward to receiving your revised manuscript.

Kind regards,

Hannah Xiaoyan Hui

Academic Editor

PLOS ONE

Journal Requirements:

"We thank North Dakota Flow Cytometry & Cell Sorting Core for services. This work was supported by the

National Science Foundation [1942471 to M.]."

"This work was supported by the

National Science Foundation, https://www.nsf.gov [1942471 to Manu]. The funders had no role in study design, data collection and analysis, decision to publish, or preparation of the manuscript."

Reviewers' comments:

Reviewer's Responses to Questions

**Comments to the Author**

1. Is the manuscript technically sound, and do the data support the conclusions?

Reviewer #1: Yes

2. Has the statistical analysis been performed appropriately and rigorously? 

Reviewer #1: Yes

3. Have the authors made all data underlying the findings in their manuscript fully available?

Reviewer #1: Yes

4. Is the manuscript presented in an intelligible fashion and written in standard English?

Reviewer #1: Yes

5. Review Comments to the Author

Reviewer #1: In this text, authors claimed a Lucflow method to test the luciferase expression in single cell and thought it is a better way than that in bulk cells. In fact, they did not examine the activity of luciferase but used immunostaining of luciferase. So, any reporter protein is suitable for this assay. It is not clear why to choose luciferase as a candidate protein. How about using GFP or other fluorescent proteins？

They found that dead cells had no detectable luciferase protein by comparing the fluorescent signal with control dead cell. It is suggested that authors can examine luciferase activity in dead cell transfected with luciferase proteins. Because luciferase activity assay is always used and quite sensitive in bulk cells.

Overall, authors still provided a new idea to analyze the cells containing luciferase proteins and validated it is feasible. If luciferase gene was knocked in a low abundant cell type, their method maybe a good choice to analyze the expression of this cell type. So authors should more deliberately reorganize their discussion.

There is no error bar in Figure 4.

6. PLOS authors have the option to publish the peer review history of their article (what does this mean?). If published, this will include your full peer review and any attached files.

Reviewer #1: No

---

## [Author Response · Author response to Decision Letter 0]

4 Sep 2023

Please see attached file labeled "Response to Reviewers".

---

## [Editor Report · Decision Letter 1]

18 Sep 2023

LucFlow: A method to measure Luciferase reporter expression in single cells

PONE-D-23-20181R1

Dear Dr. Manu,

We’re pleased to inform you that your manuscript has been judged scientifically suitable for publication and will be formally accepted for publication once it meets all outstanding technical requirements.

Kind regards,

Hannah Xiaoyan Hui

Academic Editor

PLOS ONE
---

## [Editor Report · Acceptance letter]

25 Sep 2023

PONE-D-23-20181R1 

LucFlow: A method to measure Luciferase reporter expression in single cells 

Dear Dr. Manu:

I'm pleased to inform you that your manuscript has been deemed suitable for publication in PLOS ONE. Congratulations! Your manuscript is now with our production department. 

Kind regards, 

on behalf of

Dr. Xiaoyan Hui 

Academic Editor

PLOS ONE